# Of Force? Plasticity, Annihilation and Change

Dylan Jeffrey Cree

School of Journalism, Writing, and Media, University of British Columbia, Vancouver, BC V6T 1Z4, Canada; dylancre@mail.ubc.ca

**Abstract:** Catherine Malabou's conception of plasticity as potentially having a creative or destructive form provides both philosophy and the neurosciences with a dynamic and generative concept for describing the workings and transformations of psychological, social, and material phenomena. Exploring the dynamism of Malabou's plasticity, I question: how is plasticity, whether as a giving or receiving form, constituted to be so dynamic? Drawing somewhat from Heidegger's account of change, I propose thinking of form as existing within a world of forces, to be a force, and be composed of force(s). The problem being, though somewhat presupposed and even alluded to in her elaborations of form and destructive plasticity, Malabou doesn't conceptualize force nor advance it as a necessity for conceptualizing plasticity. Nevertheless, developing upon Christopher Watkin's idea for engaging Malabou's plasticity relationally within a broader ecology, we come to see how, whether ontically or ontologically, force(s) appear to be what makes plasticity dynamic. As a result, in order to address the *figure of force* as being integral to form, I argue that Malabou will need to somehow transfigure her conception of plasticity. Ultimately, in my estimation, such elaboration may lead to plasticity's conceptual re-birth in the form of a mediating force.

**Keywords:** destructive plasticity; Malabou; force; form; metamorphic; ontology

## 1. Introduction

Catherine Malabou accounts for plasticity, both ontologically and poietically, as possessing dynamic potential for change. On one front, plasticity encapsulates a being's potential for change. In *Plasticity at the Dusk of Writing*, declaring plasticity's ontological status, that, at once, may mold or explode, she writes: "Being is none other than changing forms; being is nothing but its own mutability" (Malabou 2010, p. 43). On another front, her account brings focus to *form* by its material coming into being in the world—a thing's being given form and giving form—by and through creative and destructive forces. In all, Malabou provides the contemporary philosopher with a conceptual bridge that connects the theorizable to the neurosciences and/or plastic workings of the brain. Further, along with linking the material and conceptual, by the genius of her plasticity, Malabou most poignantly provokes re-thinking, alongside creative plasticity, the often negatively-considered *destructive*. For example, in the instance of the brain-damaged subject, instead of representing the dire termination of the supposed productive continuum of "normal" neurological plasticity, Malabou argues destructive plasticity is, in fact, no dead-end. Instead, destruction needs to be recognized as a potential force of and for change(s) in the subject that further, and more philosophically speaking, reveals the radical *other* of plasticity and of Being. Thus, plasticity not only entails becoming formed and re-formed, but it may also, as per inaugurating an entirely new unforeseeable form, entail the annihilation of form.

For Malabou, in virtue of its being the destruction of form the annihilative (which *prima facie* defies being of a form) is to be conceived as form-giving. Put another way, destruction of form does not collapse into an abyss (of the formless or, if you will, as somehow form-denying). Destruction of form is actually the production of a new form *in* and *of* the absence of form. By the logic of this thinking, plasticity is guaranteed its ontological possibility—the precondition that articulates the being of form as a becoming

that is (the precondition) of what becoming isn't (a form)—that ensures a potentially unlimited metamorphism of plasticity. Thus, concerns over destructive plasticity being an absurd proposition, of whether or not the annihilative has a form, are hereby addressed by plasticity's negative precondition. That said, and this is where Malabou's thinking becomes problematic: in accounting for the very possibility of destructive plasticity as arriving from *the nothing*—the indeterminate yolk of Being—it is somewhat unclear how Malabou distills plasticity's (form of) mutability from (its) effect and affect. Whether by the creative or the destructive, Malabou scantily elaborates on what comprises plasticity's capacity for change. Through and through, she limits the discourse on plasticity, and, more specifically destructive plasticity, to describing the being of change as its own doing and becoming. Simply, Malabou presents plasticity as being an entirely form-governed process (of being and becoming) that, as such, is in need of a description that addresses the constituting elements of what forms or can form. Thus, I contend that her account is troublingly shy of detailed insight into what, either phenomenologically or ontologically, constitutes destructive plasticity (be it push, pull, or, as per plasticity's annihilative potential, shock, and blast), and of what its constitution may entail. I see this as a problem because, in not accounting for how plasticity may be destructive or creative, the concept of plasticity is somewhat disconnected from the very dynamic processes (that ultimately give and receive form) active in change. Plasticity, although I believe is unquestionably integral to being, reads as abstracted from quite possibly non-plastic transformative impetuses of life and world. Understandably, Malabou's ontology of the mutable avoids second order knowledge claims governed by tenets of reason and causality, but still, even for considering the most abstract conception of change the question arises: doesn't change happen or show? Thus, I inquire, isn't change felt or experienced as force(s), and against other forces, within the existential meaningless at-play of resistances that take form *vis a vis* plasticity?

From Malabou's account of destructive plasticity, we understand destructive plasticity to show its metamorphic capacity rather dramatically most unexpectedly, unreasonably, and violently. —Being of the void of what isn't, of what form isn't (but that which is of the potential of another form—effectively, that is *of* plasticity), destructive plasticity represents a radically contingent explosion of form. Clearly, plasticity, in virtue of being of the nothing, entails an explosion of form. Further, for Malabou, an explosion of form—destruction—serves as a proxy, or, possibly a conduit, for force. Even so, we lack a sense of what may existentially comprise explosions/destruction. Integral to a *being's* mutability, what forces are at play? Forces, we may surmise, that aren't simply resultant phenomena secondary to the doings of form, but are also *of* the nothingness of being. Although Malabou may infer force by a generalized determination of destruction, as well as, through notions such as capacity, tension, and resistance, she does not explicitly conceptualize force, or, at least, specify why and how to consider force in relation to form and change.

Of course, there's little to support the idea that Malabou's somehow a closeted adherent of Aristotelian metaphysics, and further, as we shall discuss, it's doubtful that she defers to Heidegger's ruminations on *Kraft* and *Kraftlosigkeit*. Effectively, there's not much that can be drawn from her texts, explicitly or even subtextually, that suffices as a viable and coherent working definition of force. Elaborating on force as being integral to form would appear to be in step with Hegel's *Phenomenology of Spirit*, from which Malabou derives her account of plasticity. In fact, in *Phenomenology of Spirit* Hegel not only posits a relational notion of force that shows through opposition with other forces, he actually goes so far as to articulate the necessity of force for form, even to the point of substituting form as a force: "What appears as an 'other' and solicits Force, both to expression and to a return into itself, directly proves to be *itself Force*; for the 'other' shows itself to be as much a universal medium as a One, and in such a way that each of these forms at the same time appears only as a vanishing moment" (Hegel 1977, p. 83). Extending Hegel's logic to Malabou's plasticity we may ask: What would it mean to say "form is force" and "force is form"? If anything, I believe there's a need within Malabou's ontology of plasticity to elaborate on plasticity as being of an implicit expenditure of forces, and/or, as possibly

being within a general economy of forces. As such, if *force* is to be reckoned with as a *force majeure* at play in Malabou's analysis, apt to the volatile and incendiary nature of plasticity, we potentially detonate a series of explosive questions dormant within the very ontology of plasticity. Accordingly, in a most plastic manner, insomuch as such may even prompt Malabou to re-articulate destructive plasticity, more so than a concept of force as latent feature in Malabou's texts, I argue the need for Malabou to account for a force such that it necessarily implodes, expands and contracts her general economy of plastic.

Ultimately, the force(s) of destructive plasticity relates to the broader ontological play of Malabou's account of change and transformation (that is of *the nothing*). In effect, using destructive plasticity as an entry into discussing force, I bring attention to the necessity of the *figure of force* for Malabou's metamorphic conception of the "spontaneity and receptivity" (Malabou 2005, p. 186) of plasticity. I contend that force is important for Malabou, not simply because it is a figure of the being and becoming(s) of plasticity, but because the registering of and reckoning with force(s) potentially elaborates plasticity's dynamism as, to put in terms that theorist Christopher Watkin (to whom I will later turn to for examining the necessity of the figure of force for plasticity) might characterize things, an epigenetic socio-biological being and becoming. In this sense, perhaps plasticity, as the giving and taking of form, need then be re-thought as a mediating force.

## 2. Two Economies of Change(?)—No Costs Spared for Destruction

Acknowledgment of the role of destructive plasticity allows us to radicalize the deconstruction of subjectivity, to stamp it anew. This recognition reveals that the power of annihilation hides within the very constitution of identity, a virtual coldness that is not only the fate of the brain injured, schizophrenics, and serial killers, but is also the signature of the law of being that always appears to be on the point of abandoning itself, escaping. An ontology of modification must shelter this particular type of metamorphosis that is a farewell to being itself. A farewell that is not death, a farewell that occurs within life, just like the indifference of life to life by which survival sometimes manifests itself. (Malabou 2012, pp. 37–38)

In the quote above, beyond responding to Freud's account of the analyzable sustaining subject (Malabou 2012, pp. 81–91) and providing a way to "deconstruct subjectivity" Malabou advances the *annihilative* as fundamental for how she conceives of plasticity. Here, she articulates destructive plasticity as having the very depth and significance (as does creative plasticity) for making and transforming. More significantly, Malabou, although explicating destructive plasticity in dramatic contrast to creative plasticity, most crucial to her contribution to contemporary philosophy, finds a way to negotiate the destructive and the creative as one, as unified. I contend that the very integration of the destructive with the creative is of paramount importance for Malabou because it articulates the ontological condition for the (without cause or entirely unexpected) annihilative, and further, that plasticity's negation—*to be other*—is *of* plasticity's possible metamorphism. Accordingly, in this section, for staging a discussion on *force as a constitutive figure of plasticity and mutability*, I focus on how destruction represented as "the signature of a law of being that always appears to be on the point of abandoning itself" is *to be* understood for grasping, in its singularity—as being, at once, both creative and destructive—the conceptual framework of Malabou's plasticity. The point being, rather than taking the relation between destructive and creative plasticity as an ontological given, by staking out the philosophical terrain in which plasticity remolds anew (by "explosion of form") within its *own* orbit of expanding materialisms I look to piece together the reasoning for *how* annihilation has been conceived as being integral to the formation of things. To my mind, accessing the reasoning by which Malabou's new materialisms fosters a dynamic interplay of creation and destruction (in the giving and receiving of the form) clarifies the terms by which we may effectively contemplate what of the void, the nothing, from which destruction arrives constitutes destruction's form and the possibility for change. As stated above, I believe we need to

foreground the *figure of force* in order to animate Malabou's scene of change, the mutable plane of being.

In her book *Plasticity and The Dusk of Writing: Dialectic, Destruction, Deconstruction* and subsequent essay *Grammatology and Plasticity* (Malabou 2007), Malabou argues that deconstruction (primarily Jacques Derrida's form of deconstruction) can no longer be an adequate model for describing the transformative character of our material reality. In *Dusk*, she writes,

> In my view, however, writing does not have this capacity ["to incorporate the historically nongrammatological character of its supplements"]. There is in fact a power of fabrication of meaning that exceeds the graphic sign. This nongraphic supplement does not introduce a logocentric residue, but it marks the *difference of the grammatological instance* from itself which is also its twilight. Indeed, it seems that from now on plasticity imposes itself, gradually but surely, as the pervading figure of the system of the real in general. The brain's plasticity presents a model of organization that can still be described in terms of an imprint economy, but neuronal traces don't proceed as do writing traces: *they do not leave a trace*; they occur as *changes of form*. (Malabou 2010, p. 79)

As per the permutations of the neurological and the social, Malabou claims the concept of plasticity has far greater potential than the trace, or writing, for expressing and describing the dynamic possibility of material form, and ultimately, the mutability of beings. In contrast, writing is limited by its imprint economy. While writing can account for the trace as a graphic supplement, Malabou argues the "nongrammatological" exceeds writing's capacity (Malabou 2010, p. 79). Derrida's trace thus appears restricted to the scene of the graphic supplement that, as Malabou claims, is marked off from the "nongraphic supplement [that] does not introduce a logocentric residue". Conversely, plasticity entails the "power of fabrication" or, if you will, the very force for the molding of form that brings about, as opposed to the remnants of change, actual change. Accordingly, Derrida's grammatological approach represents a scheme in which only trace elements (of the inscribed)—where that which may have been present may have become absent—may be identified. Most importantly, according to Malabou, we cannot describe processes of change, within brains or any other formation for that matter, beyond what the *sous rature* chain of supplements allows. In a nutshell, Derrida's grammatology is limited to reading the actions and results of printing tools within a broad context of writing. His is a theory that can only play out as a reflexive representation distanced from things, much like the pen that writes on parchment, as things to be acted on.

Malabou, upon contending that grammatology can really only account for inscription, re-directs the conversation towards supposed precedent non-text or non-code-based processes. Conversely, Malabou identifies a way for integrating actions, constructive and destructive, within the actual make-up and being of materials. She conceives of plasticity as a dual-but-unified form. That is, by its transformativity, it shows what is expressed while being the very material that facilitates expression. Generally considered, her new materialisms eliminate the distance between object and action. By the concept of plasticity, we grasp the shape and transformation of things as being metamorphic, or, as being a generative function that engenders its own material possibility for its negation—for no longer functioning, or, perhaps, functioning otherwise. Moreover, we gain further perspective on Malabou's previously mentioned ontological claim that "Being is none other than changing forms; being is nothing but its own mutability" (Malabou 2010, p. 43). As inherently transformative, plasticity permits, in virtue of reforming and deforming, exceeding the capacity of its form. It is for this reason that Malabou argues plasticity is simply more dynamic for, and thus relevant to, understanding brain functioning, being, etc. than Derrida's account of writing (which possesses inherent limitations). Outlining her idea for a new seemingly auto-renewable model of description, be it for the brain sciences or the workings of *the social*, Malabou professes, in terms of being the "motor scheme" of

the form for what potentially takes form, the politico-philosophical significance of plasticity. Against Derrida's *ecriture* she writes:

> I realized that writing was no longer the right image and that plasticity now presented itself as the best-suited and most eloquent motor scheme *for our time* (Malabou 2010, pp. 14–15).

And, clarifying the character and value of a motor scheme Malabou elaborates:

> A motor scheme, the pure image of a thought—plasticity, time, writing—is a type of tool capable of garnering the greatest quantity of energy and information in the text of an epoch. It gathers and develops the meanings and tendencies that impregnate the culture at a given moment as floating images, which constitute, both vaguely and definitely, a material "atmosphere" or Stimmung ("humor", "affective tonality"). A motor scheme is what Hegel calls the characteristic (Eigentümlichkeit) of an epoch, its style or individual brand. As a general design if you wish, the movement of a whole is an initiating process for action or practice. (Malabou 2010, p. 14)

For Malabou, signifying the merging of material and concept, her motor scheme is the "image" by which form and function are one. Further, and apt for discussing force constituting form, in step with Hegel, Malabou's plasticity appears to engage and engender other animating and affective elements such as "energy". Although the objective of this section is to analyze the framework for reckoning *the destructive* with *the creative*, it needn't elude us that Malabou makes allusions to force. As a motor scheme, plasticity pictorializes the movement of forces and the force of movements, and more to my general point, seemingly improving upon the supposed imprint mechanism of Derrida's trace economy, implies a broader framework of forces in which plasticity's conatus (in whatever manifestation) persists, resists or may even encounter resistance. In this sense, it is arguable that Malabou's vision for plasticity (by its malleability, reflexivity, generativity, and destructiveness), as a motor scheme that articulates manifold differences of *the contemporary*, thus reads parallel with Hegel's general account of what force expresses, from moment to moment, as being the "dispersal of independent 'matters'" (Hegel 1977, p. 81). Like Malabou's motor scheme, in virtue of its being spent, Hegel's force remains unified and generative. As I will discuss later, whether, in terms laid out by Hegel or even Heidegger, force(s) shows negatively. Force isn't just a facet of what forms, rather, it is integral to how the form *is*, becomes, sustains and, specific to Hegel's *Phenomenology*, disappears.

At any rate, throughout her work, Malabou conceptually navigates the philosophical terrain of risk, violence, damage, trauma, and indeterminacy showing how destruction is integral to being and to the makings of our world. In giving play to the discursive (i.e., the written) and non-discursive (i.e., the brain) as a formative union, she re-shapes the philosophical landscape for how one may better account for *reality* as the fluctuating of either material and political manifestations that appear and disappear. As we know, Malabou is steadfast on the brain being central to engaging such transformative processes. As a result, she has taken on the monumental task of reconciling two very different realms of inquiry. Her wonderfully ambitious project to bridge philosophy and neuroscience requires, in the form of multi-lateral thinking, seeing connections between divergent concepts and methodologies, as well as, navigating vastly distinct discourses. Plasticity, along with, as we will explore, her later and very important turn to epigenesis, potentially provides a constructive conceptual model that dynamically accounts for the fluid and transformative interplay of the structural and the contingent, the biological and the social. As indicated above, Malabou believes her form of new materialisms, at once, marks both the zenith of a post-deconstructionist analytic and "the twilight" of Derrida's deconstruction and/or, in general, text-centered western philosophy. In fact, one of the main themes of Malabou's texts is that of differentiating plasticity from deconstruction specifically in order to demonstrate plasticity's superior explanatory powers over Derrida's deconstruction. That as it may be, I believe Malabou can only assume her analysis superior to Derrida's analytic—for providing "the right image" (Malabou 2010, p. 15) for "the motor scheme

for our time"—by overlooking key elements of Derrida's account of deconstruction and writing that actually extend beyond writing, that are also of the non-discursive. Focused on writing as *only* structural, Malabou glosses the violence or force of inscription and the kind of feasting, or logic of consumption upon which deconstruction is predicated. From my perspective, like Malabou, and by the same non-binary thinking, Derrida's analysis is also to be thought of in terms of destruction being integral to the formative or generative processes that constitute systems. So, although Malabou's plasticity may indeed better articulate the transformative potential of systems, as I explain, her interpretation of Derrida's system of supplementarity needs considerable supplementing.

Actually, Malabou's account of plasticity, which incorporates the annihilative within the making of new forms of things, is closer to Derrida's deconstruction and account of writing than she is willing to acknowledge—in fact, there's a fair bit more compatibility than there is a difference. Thus, I explain how Malabou's thinking about plasticity intersects Derrida's orbit of deconstruction, specifically in terms of how Derrida accounts for deconstruction with respect to *the parasite*. In attending to Derrida's thinking on the destructive parasite, I not only seek to clarify his depth of thought, but I also have an eye to how Derrida's thinking of the parasite along with his general approach to deconstruction contributes to the broader conversation concerning force and change. Effectively, by probing the reserves of Malabou side-by-side with Derrida's analyticity we may ascertain the critical elements and processes *vis a vis* forces for conceiving what may possibly emerge, change, or transform. In the next section, to explore both the force plasticity implies and how force is integral to change, again, Derrida will prove to be helpful. Throughout many of his texts, Derrida overtly grapples with how to conceive of force(s) for thinking about the deconstruction of political, juridical, and philosophical institutional order. Apt to *being's* metamorphic potential that Malabou elucidates in her text *The Heidegger Change*, Derrida's analysis of force (which challenges Heidegger's presuppositions about force and forcelessness) opens important questions regarding the notion of change.

Much like Malabou's new materialisms, Derrida's deconstruction problematizes representational outside-looking-in conceptual frameworks/models for analyzing systems. The main insight of Derrida's formulation of deconstruction is how deconstruction is an activity that is integral to the formation and structure of systems it deconstructs. In a "Letter to a Japanese Friend" Derrida writes,

> Deconstruction takes place, it is an event that does not await the deliberation, consciousness, or organization of a subject, or even of modernity. It deconstructs itself. It can be deconstructed. [Ça se deconstruit.] The "it" [ça] is not here an impersonal thing that is opposed to some egological subjectivity. It is in deconstruction (the Littré says, "to deconstruct itself [se deconstruire] . . . to lose its construction"). And the "se" of "se deconstruire", which is not the reflexivity of an ego or of a consciousness, bears the whole enigma. (Derrida 1985)

Malabou, however, seems to disregard the import of the nature of deconstruction's embeddedness. We see in texts such as "The End of Writing? Grammatology and Plasticity", that Malabou casts writing/deconstruction only as a discursive act or strategy (independent from the institutions one may deconstruct) for which its modifiability is made possible by plasticity. Even so, deconstruction is very much a part of the systems it may dismantle which, for Derrida, informs the deconstructive strategies for breaking apart the implicit absolutes of a structure. Very much like Malabou's plasticity, deconstruction involves grappling both with destructiveness integral to the construction of systems, as well as, the kinds of potential for the transformation of things that such an integrated-type (i.e., constructive/destructive) force may harbor. To better understand the generative and degenerative forces active in and active as deconstruction (its motor scheme if you will) I turn to Derrida's claims regarding *the parasite*. In fact, for Malabou to rightly account for Derrida's "imprint economy" I believe she needs to contend with the role and specter of the parasite. That is, for how the grammatological harbors the non-grammatological, or, to put another way, for how rules are conditioned by the unruly. In an interview with Peter

Brunette and David Wills Derrida responds to a question about his *The Post Card* and its relation to technology:

> all I have done, to summarize it very reductively, is dominated by the thought of a virus, what could be called a parasitology, a virology, the virus being many things. I have written about this in a recent text on drugs. The virus is in part a parasite that destroys, that introduces disorder into communication. Even from the biological standpoint, this is what happens with a virus; it derails a mechanism of the communicational type, its coding and decoding. On the other hand, it is something that is neither living nor nonliving; the virus is not a microbe. And if you follow these two threads, that of a parasite which disrupts destination from the communicative point of view—disrupting writing, inscription, and the coding and decoding of inscription—and which on the other hand is neither alive nor dead, you have the matrix of all that I have done since I began writing. (Brunette and Wills 1994, p. 12)

Derrida's point is that the parasite, as an agent of chaos and control, is integral to the formation of texts, systems, and rule-making. Texts are not simply a limited series of imprints in which, once the routines for conveying messages between sender and receiver become predictive, the inkwell of supplementarily will run dry. In other words, Derrida's "imprint economy" doesn't collapse or end with the last trace. Transformations of and within texts are always and need to always be occurring. Phrased another way, the investments made within this form of the economy are predicated on the very processes for the exchange and for making adjustments to unexpected, and perhaps imagined, threats.

In effect, the parasite, as an unpredictable threat and direct challenge, reinvigorates and retools *the text*, the system, etc. Still, although a parasite may be cast as a threat from the outside, its threat and manner of consumption cannot simply be determined as external. As Michel Serres points out, *parasitism* is structural. Or, as he puts it: "parasitism is an elementary relation; it is, in fact, the elements of the relation."[1]—Parasitism gives form to what becomes the relation between a host organism and a guest (perhaps unwanted) organism that feeds upon the host. The point is, in so many ways, the parasite's threat of destruction is "internalized" by a system. It is part and parcel of a broader always altering system of forces that are integral to revealing a particular system's facility. In other words, for a system's capacity to transform and show other potential, for all intents and purposes, in an expanded and divergent capacity.[2] By the above quote, accounting for the parasite in terms of *communication*, Derrida registers the atemporal dimensions of the parasite's reign of terror. Though technologies are designed to anticipate the potential threats of a virus, the endless micro-warfare within the formation of a system entails a fight with what is both, as Derrida indicates, *historical* and *non-historical* (perhaps ontological) forces. As he declares, a virus is "neither alive nor dead". A virus, at once, within and not within history—as if from some never-arriving future—may impact the future course of a system. In the "Rhetoric of Drugs", Derrida provides an outline of the parasite's logic of feasting that guides his deconstructive strategies. He tells us how deconstruction, by an economy of devouring, is active as a parasite subject to the *terms of parasitism* and subsequent alterations to how consumption may occur.

> The bad pharmakon can always parasitize the good *pharmakon*, bad repetition can always parasitize good repetition. This parasitism is at once accidental and essential. Like any good parasite, it is at once inside and outside—the outside feeding on the inside. With this model of feeding, we are very close to what in the modern sense of the word we call drugs, which are usually to be "consumed". "Deconstruction" is always attentive to this indestructible logic of parasitism. As a discourse, deconstruction is always a discourse about the parasite, itself a device parasitic on the subject of the parasite, a discourse "on parasite" and in the logic of the "super-parasite". (Derrida 1995, p. 234)

In effect, Derrida's account of parasitism as a feeding machine—that through disorder prompts a system's particular kind of formation, reformation, and transformation, as part of *difference* (in the making of the mark)—parallels, ontically and ontologically, the creative and destructive motor scheme of Malabou's plasticity (how the form of its matter and the matter of its form transforms). The point being, Derrida's imprint economy may also be, rather than just representing fading investments in *the once-inscribed*, invaluable for *what* we may assess as sociopolitical formations. —The very formations Malabou prompts us to think about in her book *What Should We Do With Our Brain*?

As well as how we may think about sociopolitical formations, there are certainly more conceptual parallels between deconstruction and plasticity. Much like plasticity giving form to the future, the parasite also *engenders* a material reality. That is, similar to how Malabou conceives of plasticity as an adaptive "motor scheme" active in the complex workings of our contemporary social and political reality, Derrida's trace is both carrier and force within a renewable economy of consumption. For Derrida, the parasite/system relation (which suggests that systems are also parasitic) possesses the adaptability, impressionability, and resiliency of plasticity. In all, it appears, they both account for *the destructive* non-binarily, as being integral to the form of things and, in some manner or another, generative. However, along with describing how things are integrated, Derrida also probes how the parasite/system relationship may be constituted as and by always varying modes of consumption. This, in turn, requires having to account for a world of always different and changing forces and formations. Thus, what is noteworthy here as a difference between deconstruction and Malabou's plasticityis how Derrida articulates the workings of systems. Consistently, Derrida necessarily engages parasitism (in all its mutations, and influences against, besides, within, and by parasites) as mechanisms of exclusion active through rules that articulate, conceal and/or execute subjugation and exploitation. Apt to parasitism's seemingly destructive attributes "subjugation" "consumption", "violence", threat of violence", "restriction", etc. deconstruction exposes the impact and influence, the very forces that the encoding and the encoded entail. To be clear, just as we see from a "Letter to a Japanese Friend", deconstructive thinking is, in some manner, always already infected by previous systems of thought, as well as, being a part of, whether past or present, a broader economy of forces.

In fact, throughout his texts, Derrida's writing displays a complex engagement with force, especially in terms of how one may conceive of force. For his colloquium address "Force of the Law", he goes so far as to say, "I've always been uncomfortable with the word force, which I've often judged to be indispensable" (Derrida 1992, p. 7). That said, the point being, while identifying the force and form of parasitic constructs/relations, Derrida reminds us that not only is content-making a biased or selective activity but so are the mechanisms for forming systems (which may very well, for whatever reason, make certain phenomena, events, experiences obscure and virtually untraceable). For Derrida, force(s), *aliquo modo*, is a constitutive feature of *the social*, our concepts, and the approaches we deploy in analyzing and describing the world. On the flipside, Malabou however, appears resolved to delimit her thinking of the constitution of plasticity by the self-affirming economy of its malleability. That as it may be, how Derrida wrestles with the indeterminacy of systems (that are within a broad ecology of relations) as engendering the figure of force may provide us with insight into how we begin to engage the *economy of forces that are active as plasticity's giving and receiving*.

### 3. The Force(s) of Changes to Come

In passing from one motor to the other, from one energetic device to the other force simultaneously loses itself and forms itself differently, just as the metamorphic crisis frees a butterfly from its chrysalis (Malabou 2008, pp. 73–74).

Even though, for the most part, Malabou and Derrida's thinking on the role of *the destructive* are grounded in the same kind of non-binary thinking, their accounts of how to characterize the changes or transformations that destruction may yield appear to diverge.

For both, in one manner or another, destruction is integral to the organizational or formative being and becoming of things. As per plasticity, for Malabou, through violence and explosion destruction, in virtue of absence or negation, renews form with the emergence of another kind of form. Similarly, for Derrida's deconstruction, destruction (whether at the peril of or for the gainful purposes of a system), through the parasite as a figure of chaos and uncontrolled threat, often shows by anticipatory thwarting mechanisms and the replacing or altering of rule and code with different rule and code. That said, to discuss how, for Malabou, change is brought about by or is inherent in *the destructive* requires us to now transition from examining how destruction is integral to things to explore, given the indeterminacy of violent and destructive forces at play within the workings of systems, her thinking concerning the mutable (the ontological condition for change) which, to my thinking, requires directly and thoroughly contending with the figure of force.

In *The Future of Hegel* Malabou declares, "By 'plasticity' we mean first of all the excess of the future over the future" (Malabou 2005, p. 6). She is determined to keep the discussion and her economy of temporal horizons within the confines of supposed lived experience. While Malabou drills further into the reserves of materialisms and deepens her philosophical commitment to ontology Derrida, it would appear, turns toward the spectral. By contrast, Derrida's conception of the parasite lends to speculation that the parasite may enact both *an excess of enclosure* and *determination of what will come* through the looming presence of 'that which must be preceded' (which, of course, may be entirely indeterminate).[3] Highlighting her divergence from Derrida, Malabou states her main objective for how she characterizes ongoing transformation.

> I am just trying to show how a being, in its fragile and finite mutability, can experience the materiality of existence and transform its ontological meaning. The impossibility of fleeing means first of all the impossibility of fleeing oneself. It is within the very frame of this impossibility that I propose a philosophical change of perspective that focuses on closure as its principal object. (Malabou 2010, pp. 81–82)

For Malabou, plasticity provides an account of *the real* that expresses the possibility of change all within materialist terms. The transformative nature of being—core to the neurosciences, philosophy, and political life—may thus show through the variability, malleability, and destructibility of her concept of plasticity. However, even though Derrida's thinking (regarding the effects of the parasite or deconstruction *per se*) entails a radical indeterminacy that, because of his turn to the impossible, may not be entirely commensurate with a scientific view of reality nor an ontology of what is and isn't I contend it is unclear how much different from Derrida's notion of deconstruction Malabou's concept of plasticity actually is. Just like plasticity being form-inherent, any deconstructive discourse is implicated in what and how it may deconstruct. Recall from earlier when Derrida states: "Deconstruction takes place, it is an event that does not await the deliberation, consciousness, or organization of a subject, or even of modernity. It deconstructs itself. It can be deconstructed. [Ça se deconstruit.] The "it" [ça] is not here an impersonal thing that is opposed to some egological subjectivity. It is in deconstruction . . . " Further, if, as I have suggested, deconstruction is governed by a principle of feeding then, deconstruction is *always already* potentially transformable. To restate, given Derrida's reflections on the possibilities of the parasite as an incessant voracious contagion, it would appear his account of consumption is also, like plasticity, very much at work in a material sense.

It seems to me that the difference between Malabou's version of transformation and Derrida's notion of how transformation occurs has more to do with their *ideas* about what change entails than whether a change will actually occur. Derrida appears to have a functionally indeterminate notion of change or mutability. He does *not* account for how things, systems, rules, etc., may be altered, adjusted, or come to show as different in the process of transformation. Rather, we are uncertain as to how Derrida may characterize a system's response to a threat or actual destruction by a virus. The following questions simply go unanswered. Is a response or apparent change within a system anything other

than a chance turn and mutation within the cycle of parasitism, of the host being revealed as a guest? By what design or form or, apt to the focus of my analysis, force does any transformation occur? For Malabou, change would appear to, regardless of the basis of the change, entail reading adaptation into transformation. As touched upon earlier, whether disabled, damaged, or destroyed the new or changed form of things sustains by and as another mode of functioning, of what maintains in virtue of its plasticity. For example, brains that may be damaged merely form different kinds of brains. Malabou's functionalizing of destruction in no way parallels the aleatory character of Derrida's parasite. She makes sense of the obliterative. Destructive plasticity may completely destroy roles and goals and *have that possibility* as its role, goal, and/or outcome. Thus, for Malabou, there can be no measure of indeterminacy to the form that deforms. Any conceivable indeterminacy would, by design, take a determinate form. Accordingly, we would want to know: what is it about plastic, whether conceptually or materially, that permits such a radical functionalizing of the destructive? What elements, forces as it were, might be at play?

In her recent text *Before Tomorrow*, an analysis of Kant's notion of epigenesis, Malabou's plasticity actually may be interpreted as being in agreement with or in closer proximity to the aleatory character of Derrida's parasite than as I just explicated. Going against a tradition of Kant interpreters and critics that read the transcendental in terms of foundational solidity, Malabou, reckoning with the relation between the structure for thinking and cognition with that of experience, re-orients analyzing the transcendental as a mobile threshold, a passage, and contact point between life and reason. Given such, perhaps plasticity's form-giving and form-given may, in fact, be now explainable in terms of a globally-thought transformative epigenesis. Malabou writes

> The epigenetic transformation of necessity and causality, starting from reason itself, reveals that contingency derives less from a possible modification of the laws of physics than from the existence of different levels of necessity or lawfulness in which physical necessity is but one dimension. The epigenesis of reason: it is, therefore, important to understand the genitive in the phrase as a subjective genitive. It is indeed about epigenesis, that is, about the gestation and embryogenesis of reason itself. Throughout the itinerary of the three Critiques, the reason is transformed, and the last stage of the trajectory organizes the reflection of reason on this transformation a posteriori. The transformation does not come from outside, nor is it linked, or no longer only linked, to the type of object examined; it responds to a fundamental internal demand of reason, one that is already present, as a germ in the first Critique, and already manifest in the idea—unacceptable to many, even if it is incontestable –of a certain changeability of the categories. ([Malabou 2016](), p. 173)

In this way, Malabou, like Derrida's *always already* at play deconstruction, accounts for what takes form—of what grows and transforms and evolves—as it *will have formed* and does form. More significantly, Malabou's epigenesis may in fact represent a highly dynamic way to elaborate on plasticity as part and parcel of broader ecological processes. Malabou scholar Christopher Watkin (*French Philosophy Today: New Figures of the Human in Badiou, Meillassoux, Malabou, Serres, and Latour*) over Malabou's seemingly restricted or atomized ([Watkin 2016](), p. 127) version of plasticity, proffers a version of plasticity as a globalized interactive process. He elaborates:

> The self is not the product of epigenesis, not an object standing at the static endpoint of a process of interpretation, nor again a particular configuration of neuronal synapses that encodes such and such a set of memories. To be consistent with Malabou's incisive reading of Hegel we must hold that the self is not a particular configuration of meanings or synapses but rather the process of tension, resistance and plasticity that transforms those epigenetic and hermeneutic connections. The self is epigenetic and hermeneutic, rather than simply an encoded product of epigenesis or a hermeneutic interpretation. . . . an important advance

> towards an ecological notion of the self beyond the subject/object dichotomy is
> that, as radically distributed, the eco-synaptic self is a possession neither of the
> individual—such that I have autonomous and unimpeded sovereignty over my
> own identity, as in the modern concept of the self found in Locke and Descartes—
> nor of the wider society or community—such that my identity can be defined
> for me, against my will and my personhood circumscribed or taken away by
> an instance outside of me—but rather the self eludes the proprietorial claims of
> both radical autonomy and radical heteronomy. The self is in an irresolvable
> tension between possession and dispossession, not an appropriable substance
> but a changing series of synapses both biological and cultural, a bio-cultural
> epigenetic efflorescence that cannot be atomised and therefore owned, either by
> itself or by another. (Watkin 2016, p. 139)

Although focused on *self* and *identity*, Watkin's take on epigenesis, as warranting a
general economy for the constitution of identity, provides critical insight into the ontological
status of plasticity as "the structure—the metabolism—of the becoming of being itself".
(Watkin 2016, p. 90). As per destructive plasticity, true to Malabou's rejection of the
metaphysical belief that self, life, being, and becoming are anchored by or as an enduring
substance, Watkin articulates the importance of epigenesis for animating *the transformative*
as a process or a becoming-form that occurs within or in relation to a host of identifiable
(and likely unidentifiable) forces encapsulated by what Watkin describes, in concert with
the plasticity of things, as "tension" and "resistance". Just as he describes the at-play
processes of the self as a "distributed network of relationships" (Watkin 2016, p. 138), the
"biological" and "social" comprise and are a part of an always-transforming network(s) of
forces or forms that give rise and way to other forces or forms.

Actually, Watkin's analysis provides us considerable traction for there being a need
to identify the *figure of force* as a feature of Malabou's account of plasticity. At various
moments, in his text, he emphasizes the existential push-pull and gravity that animates
both her thinking and plasticity as a concept:

> In the same way that, for Malabou's Hegel, humanity cannot be understood
> simply as the giving of form or simply as the receiving of form but must compre-
> hend them together in their resistance to each other, so also here it is in the very
> antagonism between cognitivism and antireductionism that Malabou seeks to
> unfold the unique contribution of her own thinking. A reasonable materialism is
> one that takes account of 'the dialectical tension that at once binds and opposes
> naturalness and intentionality', a tension which Malabou expresses and explores
> in her elaboration of 'a supple and—so to speak—plastic materialism'. (Watkin
> 2016, p. 97)

Specifically, by the tropes "tension" and "resistance" Watkin guides us through think-
ing about the giving of form and receiving of form of plasticity, both dialectically and as a
*chance* dynamic. As just mentioned, to get past what he sees as Malabou's atomistic view
of plasticity, drawing on the philosophical implications of Malabou's recent turn towards
epigenesis in *Before Tomorrow*, Watkin interprets the very capacity of plasticity as being
engendered by and through the off-script dynamics of broader socio-biological processes.
As for the issue at hand, his analysis provokes thinking about how receiving, giving and
the annihilation of form, apart from having a generalized plasticity, is a multi-valent differ-
entiated process in which and by which divergent capacities show. Accordingly, we're led
to the question: by what economy of affordances and disturbances does plasticity occur
and transform? Or, more bluntly, what are the kinds of *forces* that inhere in the very motor
drive of plasticity? As it stands, given the general uncertainty over how forces may be
at play throughout the course of Malabou's myriad elaborations of plasticity, we really
only have a no-calorie version of plasticity that does little to satisfy the phenomenologist's
hunger. That said, by riffing off of Watkin's development of epigenetic plasticity (which
explodes the nuclear version of plasticity as well as, in virtue of such atomism, the notion
that *all is plastic*) I see that elaborating on the giving, receiving, and destruction of form in

terms of the figure of force is key to both a phenomenologically dynamic and dynamically transformative account of plasticity. Accounting for force potentially knits or melds together the transformational processes of plastic's becoming with always-changing, whether fractured or cohesive, material and social relationships (i.e., within the ecology of things, beings, and becomings).

Although it seems by Watkin's explication of things that we may surmise force is implicitly a part of the particular becomings of whatever form, the more penetrating question arises: what elemental significance might actually be attributed to the figure of force for the very being of plasticity? Essentially, by this question, I'm inquiring into how we may articulate the figure of force as if it were a primary aspect of the action and activity and activation of form, of what, integral to forming, constitutes plasticity's plasticity—its malleability, its destructiveness, in all, its mutability. By my thinking, resistance and tension are not only, as Watkin devises above, processes that accompany plasticity for what may or may not transform or come into being. Resistance and tension, I believe, are also representative of the kinds of processes or forces integral to plasticity. However, so that we don't lapse into a discourse presupposing a producerly mechanistic paradigm for plasticity, how we think about force needs to be mapped out.

At the start of this section I quote from a passage of Malabou's *What Should We Do With Our Brain?* in which she unpacks Henri Bergon's vitalist account of motion, specifically, the violence entailed in the "transformation of one motor regime into another" (Malabou 2008, pp. 72–73). Malabou, describing the energetic explosiveness of the relation between brain processes and thought processes (or, more specifically, the synaptic dynamism that inheres across neuronal firing and mentation) brings our attention to the ontological necessity of force as a figure integral to the transformation that, at once, parallel to the metamorphing of a butterfly becoming liberated from its chrysalis, "loses itself and forms itself differently" (Malabou 2008, pp. 73–74). Taken at face value and not as a flighty metaphor, force, apparently acting upon "itself" and as if in isolation from other forces, both negates what it does and becomes another force. Here at least, for Malabou, it appears force, as if it were a form, behaves most plastically. Accordingly, it appears there's a need to confront one of this paper's underlying questions: for Malabou's plasticity, are we to infer that force is interchangeable with form? That aside, what's more at issue is: for Malabou, how is it that force needs to be thought of in order for *it* to undo *itself* and do *itself* differently while maintaining what it *is* or *does and doesn't do*? In other words, what ontological presuppositions are at play for her account of plasticity? Here, considering how Heidegger gets us to think about the figure of force by a non-relational negativity may be of help.

As mentioned earlier, by Hegel's thinking we understand that force—by recession or diminishment of potency—shows itself as a graduated presence of a *thing* losing force. Heidegger, however, as per *Aristotle's 'Metaphysics θ 1–3': On the Essence and Actuality of Force*, eschewing a mechanistic/scientific version of force (that posits force as an isolable energy-type), grapples with the figure of force existentially, as a fluid and change-conducive element nested within the dynamic interplay of our world. So, in thinking about the connection between force and plasticity, I propose we not only need to account for force, and even the forceless, in the dialectical sense to which Hegel brings our attention (in which he accounts for force in terms of disappearance), but we need to, by force's very being and, most apt to our discussion of Malabou's conceptualizing of plasticity, do so specifically in relation to 'change.' Thinking about the relationship between force and change for Aristotle Heidegger writes:

> For what does it mean that *dynamis* is the form-out-of-which, which implicates into its own realm that which in itself is able to bear? This indeed says only that force, on the basis of its essence, first provides *a possible site for a change* from something to something. To say that what can endure is exposed to something which works it over means: something like change is already and necessarily signified in this reciprocal relation, both what permits being formed into shape as well as the forming production. (Heidegger 1995, p. 97)

For Heidegger, force shows through change whereby force, at once, enacts change and is *of* that very instance of change. Effectively, while acknowledging force's potency in things, Heidegger slips assigning a *causa prima* or origin to force. Avoiding a causal account of force is doubly important. Not only does Heidegger immerse force within the 'to and fro' of the world, but he also ontologically reconciles the existential gravity of force. For Heidegger, force is to be thought of in terms of presence and absence, or, by its *own* potency as well as its *own* unquantifiable impotence by which (and not as the result of an expenditure, which would be in step with Hegel) an unforced or 'forcelessness' inheres in the very being of force. Here, Heidegger brings our attention to what is meant by forcelessness in his translation of Aristotle's formulation of *dynamis*:

> unforce (forcelessness) and consequently also the 'forceless' is a withdrawal as what lies over and against *dynamis* in the sense developed; hence every force, if it becomes unforce, that is, as unforce is in each case in relation to and in accordance with the same (with respect to that by which a force is a force, every force is unforce). (Heidegger 1995, pp. 91–92)

And then later Heidegger expounds.

> This negativum [referring to un-force] does not simply stand beside the positive of force as its opposite but haunts this force in the force itself, and this is because every force of this type according to its essence is invested with divisiveness and so with a 'not'. (Heidegger 1995, p. 132)

Accordingly, for Heidegger, as per his interpretation of Aristotle, force is, at once, robust and, by such, is entirely empty (of its force). In its absolute positivity, potency, or presencing, force is (also) nothing. In her book *Force from Nietzsche to Derrida* Clare Connors helps to navigate Heidegger's analysis of force which, as she points out, is integral to things and beings, or, what has and takes form. Connors writes:

> [For Heidegger] Force is directed and oriented: not a general, logical, or
>
> a priori possibility, it is always a possibility for a particular thing, and thus is guided in advance by that particularity. (Connors 2010, pp. 54–55)

As Heidegger sees it, force may be situated, but, as Connors also makes clear, by its own discreteness and possibilities, "force always has an intimate relationship to a how-not-to". (Connors 2010, p. 55) This means, for Heidegger, force's dynamism is not by doing or exhibiting a thing's capability, alterability, or impact but by a thing's not doing. In essence, Heidegger's ontological account of force gives cause to observe how it so-readily aligns with Malabou's elaboration of plasticity:

> A process of formation and of the dissolution of form, plasticity, where all birth takes place, should be imagined fundamentally as an ontological combustion (déflagration) which liberates the twofold possibility of the appearance and the annihilation of presence. It is a process which functions on its own, automatically. As such, it comes out of nothing; as such, it is the bearer of the future, if it is true that the future, by definition, comes from nowhere. (Malabou 2005, p. 187)

Force and plasticity appear to engender the same potential, whether destructive or creative, *as* and *for* their coming into being and/or presencing. Just as Heidegger states that "force, on the basis of its essence, first provides *a possible site for a change* from something to something" (Heidegger 1995, p. 97) Malabou claiming plasticity, in virtue of its existential groundlessness, "is the bearer of the future" doubles down on the ontological primacy of plasticity's giving nature. Parallels aside, there are some problems nested in Heidegger's figure of force that Malabou may not want her concept of plasticity associated with. Namely, Heidegger's move to explicate unforce as predicated on force installs the bias of *being* as presence. In effect, forcelessness is but presence's negation whereas, by the precondition of spatial governance indicated by forcelessness haunting or lurking within force, force cannot be forcelessness' presence. Accounting for forcelessness as such indicates no mere transitory or peculiarly asymmetrical feature of force, rather, for Heidegger presence is a

generalized condition of being. Here, Connors is instructive in how she parses the implicit hierarchy of the force/forcelessness binary:

> Heidegger, we saw, thought about force as the origin of change. We saw how this origin was already divided, both ontological and ontic, and including both the changer and the changed. Force as archē metabolēs was a sort of trace, a presencing through withdrawal. However, there was a residual sense in Heidegger—tied in with his strong language of authenticity and of resolution—that while unforce was, as it were, the gauntlet force ran, it was still, no matter how minimally, a secondary and empirical loss which befell a force otherwise sufficient in its powers. (Connors 2010, p. 113)

By Heidegger's exploration of force/forcelessness (in relation to change) I contend Malabou's account of plasticity is up against questions concerning how force is integral to both form and change. In Malabou's *The Heidegger Change* the key terms—*Wandel*, *Wandlung*, and *Verwandlung* or respectively "change, transformation, and metamorphosis" as Malabou specifies "constitute . . . the triad of change"[4] (Malabou 2011, pp. 1–2)—shed light on the transitory nature of Heidegger's account of being. Nonetheless, as I've been claiming, Malabou, is pretty much mute concerning the significance of the figure of force, and, specifically so, in some of the texts she actually locates 'the triad' (where force clearly plays a role in Heidegger's thinking about change). Her interest in the "triad of change" is solely for mining Heidegger's texts as if they were but a reserve secreting the myriad wealth of possible varieties and variations of "change". For Malabou, at least overtly, it seems the force(s) this 'triad' possesses and may be *of* is just not a part of Heidegger's thinking. In *Dusk* Malabou states:

> The triad of change—with all its migratory and metamorphic variations and the great wealth of its differentiated structure—is the motor scheme of Heideggerian thought, hidden like a casket in the recesses of the text, an inexhaustible fantasmatic resource. (Malabou 2010, p. 31)

And yet by referring to the "triad of change" as "the motor scheme of Heidegger's thought" Malabou provokes speculation on how change animates more change. Although Malabou's account of plasticity and change loosely engages force (*vis a vis* the destructive) her description of plasticity as a motor scheme suggests more. That is, that change is both a force in Heidegger's account of being, and, that change is *of* force. Again, Watkin is helpful in how to understand the nature of Malabou's main objective: "Malabou herself frames her engagement with Heidegger as an endeavor to 'test the plasticity of the concept of plasticity even further, examining its metabolic power, its capacity to order transformation'" (Watkin 2016, p. 90). In one sense, Malabou shows Heidegger's fundamental ontology is inherently plastic. In another sense, as per Watkin's insight into Malabou's project (as per her own claim about her analysis of Heidegger) we understand that (as per epigenesis) the "itself" of plasticity's metabolism—the processes for "the becoming of being itself" (Watkin 2016, p. 90)—is a becoming that always already occurs at and as the confluence of transforming processes, be they social, biological, etc. Consequently, force is no mere word substitute or illustrative metaphor for capacity or power. Force is substantive, in that, not only does Heidegger contend with force/forcelessness in terms of change (of which Malabou misses its mention), but, that the figure of force is seemingly very integral to Heidegger's account of change. That said, does it follow that Malabou needs to take up Heidegger's lead regarding force and change? Even if it were so, how would Malabou then account for or qualify force? In the end, what subsequent impact its conceptualization would have on Malabou's plasticity may be altogether another matter. Perhaps it's prudent that Malabou is shtum on the significance of force. Maybe Malabou recognizes the pitfalls inherent in Heidegger's account of force/forcelessness and considers it wise to avoid any sustained talk of force.

Needless to say, Derrida again surfaces as a force within the conversation. As discussed earlier, Derrida, rather than as an adaptive process, considers how change, or

whatever change occurs, is indeterminate. In a narrow sense, he is mute over whether a system's response to a parasite is by design or chance. Philosophically speaking, however, Derrida systematically evades commitment to accounting for change as that which is a creation of being. Thus, for Derrida, Heidegger's being and/or Malabou's plasticity (as when Malabou claims "being is change"), although maybe harboring the possibility for change, cannot actually ensure change. Even *being* or plasticity *itself* being *change* does not guarantee that change occurs. Rather, it would appear, for Derrida, construing being as change further elaborates and conceals the fiction of an inaugural presence. Here, I believe Malabou's response to Derrida would be that 'the plasticity of plasticity addresses this lack of guarantee in that, ontologically considered, plasticity *is* the lack of guarantee'. Still, for Derrida then, the question might be: what ensures the plasticity of plasticity as the lack of guarantee? Derrida's deconstructionist intuitions resist the notion that plastic's forming and bondability, the very change that Malabou declares synonymous with being, that, as the account goes, is and occurs as and from the nothing, can be determined or ascertained as a creation (whether considered destructively or creatively) of being. Considered another way, there is no bridging ontological expression by which force, as per the indefiniteness of (its) forcelessness, is indeterminate as a determination of change. Instead, for Derrida, force considered by Heidegger as the archē metabolēs—the primordial thrusting or the changes in and into form which happens 'now', or, at every moment—tacitly confirms a creation merely derived from a particular logic, a fictive deliverance, an immaculate conception.

Ultimately, I don't consider Derrida's questioning of force and change as undermining, or, even stymieing Malabou's ontological premise of plasticity as change so much as it may prompt Malabou to transform her own account of being as change/mutability, as a giving and receiving of form that is invested with giving and receiving that are at once integral to and distinct from form.—Whereby, as form, plasticity may need to be endlessly re-described in terms of force, or if you will, of form's other. As such, Malabou's claim "By 'plasticity' we mean first of all the excess of the future over the future"[5] (Malabou 2005, p. 6), apart from reinforcing her expanding new materialisms, starts to bear considerable resemblance to Hegel's claim "Force supersedes its expression . . . Force is itself this reflectiveness-into-self, or this supersession of the expression" (Hegel 1977, p. 83). Invariably, given Malabou's commitments to an unbridled expenditure of possible futures, provocative questions of the form will arise: Is force plastic? And, is plasticity not only part and parcel of force, but also a mediating force? To those concerned with preserving plasticity in its absolute abstractness as an uncontaminated process that's consistently and only plastic, the latter question may imply that I propose, by rendering it a form of force amongst non-plastic forces, a dilution of plasticity or an undermining of the *very force* of plasticity as singular and homogeneous. That as it may be, apt to the *inherent* plasticity of plasticity, by introducing force into the mix, haven't we just renamed what plasticity is and does as a kind of medium that is, by force of form, at once pregnant and impregnating with form? In other words, in keeping with Watkin's socio-biological/epigenetic interpretation of plasticity, isn't plasticity, given the potentialities of its omni-variable malleability, always already diluted by implied force(s) and forcelessness? For Malabou plasticity, as per the conceptualized infinite vicissitudes of (its) plasticity, is inherently amenable to and capable of a subjecting of . . . while being subjected to . . . in which giving is activated by or active through force/forcelessness. That is, giving and receiving form is garnered through seemingly elemental and observable material processes.—Spring, tension, drag, and friction represent the reserve of phenomenological expressions denoting force of an explosion, implosion, expansion, contraction, etc., that tacitly imbue plasticity's forming and being formed. In this seemingly basic sense, accounting for force, and even forcelessness, serves to elaborate the very gravity, pressures, and senses of existential groundlessness, thereby expanding plasticity's phenomenological scope and deepening its ontology.

**Funding:** This research received no external funding.

**Institutional Review Board Statement:** Not applicable.

**Informed Consent Statement:** Not applicable.

**Data Availability Statement:** Not applicable.

**Conflicts of Interest:** The author declares no conflict of interest.

## Notes

[1]  See Serres (1982, p. 182). Serres developed a theory of the parasite, for which Derrida's account of the *hôte* and notion of writing has affinity.

[2]  Of course, this account is of when a system persists. It is possible the parasite entirely consumes and destroys a system. In the case of a biological system, if a system is entirely devastated and destroyed by a parasite the spoils and what remains materially of the destruction are the trace elements of what may lead to the further incarnation of things.

[3]  In other words, the *future anterior* of a system's processing that may somehow, unexpectedly, be breached in what Derrida coins *destinerrance*, or the possibility of a message deviating from its supposed goal of reaching a predetermined destination.

[4]  Expressions of change as found in, for example, Heidegger's *The Fundamental Concepts of Metaphysics* and *The Essence of Truth.*

[5]  The indeterminate "excess" of what's to come thought as a kind of boundary that overrides and resets its bounds.

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
