# Peer review of "Of Force? Plasticity, Annihilation and Change"

_humanities, doi:10.3390/h11040083_

Round 1

Reviewer 1 Report

This paper is a significant improvement over the earlier submission. Where it suffers is both in the early stage and in terms of its central argument. As such, I cannot accept it without significant changes. The most substantial error is unfortunately the central thesis: that Malabou lacks a notion of force that is needed to explain plasticity. See my attached comments for further information. I think that the current paper could merit publication with moderate changes but the thesis will at least need to be reframed or changed and the early part of the paper will need to be made more clear and consistent with what is provided from lines 121 to 748. I am suggesting that this paper be reconsidered after major revisions because the automated review submission page only allows: 1- Accept in present form; 2- Accept after minor revision; 3- Reconsider after major revision; or 4- Reject. And, given that the thesis needs to be reframed or changed (along with the first few pages), to me, amounts to more than a minor revision. However, in truth, I believe that the majority of this paper is quite good and won't require much to address. Therefore, my actual recommendation would be to accept with moderate revisions. But it will be important how the central thesis is changed or reframed since I think characterizing Malabou as utterly lacking a notion of force is a serious misreading.

Please also see the attachment.

Author Response

Thank you for your assessment. It was extremely helpful. I’ve attempted to address all the points made in the following.

From the pdf:

Review: (de)Void of Force? Plasticity, Annihilation and Change

I’ve changed the title to: Of Force? Plasticity, Annihilation and Change

- Line 13-14, Is the author here not confusing the dynamism of form with the dynamism

of plasticity?

I’ve changed “form” to “plasticity”

- Line 42, delete “its” from “its form” 

I’ve changed “its form” to “form.” 

- Line 42-43, I don’t believe the annihilative is to be conceived as a form in Malabou.

Rather, it is to be conceived of as potentially form-giving.  

I changed “form” to “form-giving.” As well, within the parentheses in the following sentence, I added “or, if you will, as somehow form-denying”).

- Line 53-54, “Malabou doesn’t provide much of an idea of what comprises plasticity’s

capacity for change.” I don’t think this is true (at least when it comes to destructive plasticity), it is destruction that provides the capacity for change quite clearly for Malabou.                                     

Now: “Whether by the creative or the destructive, Malabou scantily elaborates what comprises plasticity’s capacity for change.” 

- Line 57-58, “bereft of description that addresses the constituting elements of what – be it push, pull, or, as per plasticity’s annihilative potential, shock and blast – forms or can form” This sounds confused since you mention the very elements you claim to be missing e.g. “shock and blast”.                  

Now: “Simply, Malabou presents plasticity as being an entirely form-governed process (of being and becoming) in need of description that addresses the constituting elements of what forms or can form. Her account is shy on detailed insight into what, either phenomenologically or ontologically, constitutes destructive plasticity (be it push, pull, or, as per plasticity’s annihilative potential, shock and blast), and of what its constitution may actually entail.”    

- Line 63-64, “That is, isn’t change felt or experienced as force(s), and against other forces, within the existential meaningless at-play of resistances that take form vis a vis plasticity?” What is felt or experienced in annihilation? Your question should not be left for your reader here. If you have an argument to make on this point, make it.    

Now: “Thus, I inquire, isn’t change felt or experienced as force(s), and against other forces, within the existential meaningless at-play of resistances that take form vis a vis plasticity?”

- Line 70-71, Is not destruction a force?          

Now: “Clearly, plasticity, in virtue of being of the nothing, entails an explosion of form. And further, for Malabou, explosion of form – destruction –  serves as a proxy, or, possibly a conduit, for force. Even so, we lack a sense for what may existentially comprise explosions/destruction.”

- Line 73-74, “she does not conceptualize force, or, at least, specify why and how to consider force in relation to form”. I think there is a rather significant confusion here. Malabou clearly considers the destructive or the annihilative to be a force and she does in fact go to great lengths to explain how the destructive may give rise to form.    

Now: ”And yet, although Malabou may infer force by a generalized determination of destruction, as well as, through notions such as capacity, tension and resistance, she does not explicitly conceptualize force, or, at least, specify why and how to consider force in relation to form and change.”

- Lines 92-100, This is all very unclear. No explanation of what “force as form and/or form as force” is given, nor any rationale as to why we ought to think of them in such a way. Also, I don’t believe that annihilation is to be seen as a “feature of plasticity”, rather, it denotes a type of plasticity; the plasticity of form is revealed through annihilation in destructive plasticity.

Now:  Accordingly, in a most plastic manner, insomuch as such may even prompt Malabou to re-articulate destructive plasticity, more so than a concept of [added] force being a latent feature in Malabou’s texts, I argue the need for Malabou to account for force such that it necessarily implodes, expands and contracts her general economy of plastic.”

- Lines 112-114, quoting Malabou “Acknowledgement of the role of destructive plasticity allows us to radicalize the deconstruction of subjectivity, to stamp it anew. This recognition reveals that a power of annihilation hides within the very constitution of identity”. This quote from Malabou (and what I bolded and italicized) exposes the misunderstanding at the heart of this paper. Malabou’s referencing the “power of annihilation” explicitly shows that Malabou in fact does conceptualize the annihilative and the destructive to be a kind of force. The great error of this paper is that it’s thesis (roughly, that Malabou lacks a theory of force that is needed to further explain her understanding of plasticity) misunderstands Malabou deeply since she in fact already does provide an account of force in terms of the annihilative and the destructive—and thereby does explain the type of plasticity she is proposing as it relates to a particular kind of force. 

I have attempted to address this concern with my previous edits.

 - Lines 121—748, This is all very good but it might do better as a stand-alone paper that focusses on the tensions and intersections of Malabou and Derrida that isn’t necessarily framed by the notion of force since force is already present in Malabou in terms of the destructive.

Hopefully, I have reconciled the parts.

- Line 679-680, again, her account is not “devoid of force”     

Now: “Although Malabou’s account of plasticity and change loosely engages force (vis a vis the destructive) her description of plasticity as a motor scheme suggests more. That is, that change is both a force in Heidegger’s account of being, and, that change is of force.

Reviewer 2 Report

This is a much improved paper. The argument is much stronger and focuses on something much more central and interesting in Malabou's work.

The discussion of the relation between Derrida's parasite and the aspect of force in Malabou's plasticity is interesting, but the statement on line 284 needs an argument:

"Malabou, however, seems to disregard the nature of deconstruction’s embeddedness."

Malabou does indeed consistently refer to deconstruction, but within a different ontological context than Derrida. In "The End of Writing?" (2007), isn't what she's doing a de/reconstruction of deconstruction, establishing plasticity as the feature which allows deconstruction – and in "Can we relinquish the transcendental?" (2014), suggesting that the transcendental can be understood as plasticity as a feature of form, internal to the System? Please consider whether this should be elaborated in the paper.

Dampening the reliance on Watkin's interpretation and elaborating using Malabou's own texts could improve the text further.

Author Response

Thank you for your assessment. It was very helpful. I’ve attempted to address all the points made in the following.

The discussion of the relation between Derrida's parasite and the aspect of force in Malabou's plasticity is interesting, but the statement on line 284 needs an argument:

"Malabou, however, seems to disregard the nature of deconstruction’s embeddedness."

Malabou does indeed consistently refer to deconstruction, but within a different ontological context than Derrida. In "The End of Writing?" (2007), isn't what she's doing a de/reconstruction of deconstruction, establishing plasticity as the feature which allows deconstruction – and in "Can we relinquish the transcendental?" (2014), suggesting that the transcendental can be understood as plasticity as a feature of form, internal to the System? Please consider whether this should be elaborated in the paper.

Now (attempting to qualify my claim): “Malabou, however, seems to disregard the import of the nature of deconstruction’s embeddedness. We see in texts such as “The End of Writing? Grammatology and Plasticity” Malabou casts writing/deconstruction only as a discursive act or strategy (independent from the institutions one may deconstruct) for which its modifiability is made possible by plasticity.

Dampening the reliance on Watkin's interpretation and elaborating using Malabou's own texts could improve the text further.

I understand your concern here. However, after considering de-emphasizing Watkin, I decided he is necessary for how I set up the discussion from lines 554 and on.

Round 2

Reviewer 2 Report

It's good to see that this paper has been improved since the last round.

However, it needs further improvement before publication.

The main thesis, aim and approach are not clear enough. What's at stake, what the paper aims to do and how it approaches the problem should be explained clearly from the outset. I'm still not sure what the justification for  plasticity lacking a notion of force is, and what could be gained by exploring that.

A clear explanation of the relation between form and plasticity is needed. The first sentence in the abstract is very confusing:

"Catherine Malabou conceives of plasticity as a creative and destructive form"

– is plasticity a form? In a sense yes, but in a different sense, no, not really. The relation between form and plasticity is better handled later in the text, but clarification is needed. Plasticity at the Dusk of Writing (esp. chapters VIII, X and XI) and "Grammatology and Plasticity" could be helpful references here.

Some signposting along the way would make reading easier. Some paragraphs are much too long. Editing for readability should be done throughout.

Author Response

It's good to see that this paper has been improved since the last round.

However, it needs further improvement before publication.

To the reviewer. Thank you for your insightful reading of my essay. Hopefully, I’ve been able to address most of your concerns. Please see below and please note that the essay is now titled “Of Force? Plasticity, Annihilation and Change.”

The main thesis, aim and approach are not clear enough. What's at stake, what the paper aims to do and how it approaches the problem should be explained clearly from the outset. I'm still not sure what the justification for  plasticity lacking a notion of force is, and what could be gained by exploring that.

I have attempted to emphasize the point and purpose of my analysis by adding the following to the introductory paragraph: 

(line 89 to 97) “ I contend that her account is troublingly shy on detailed insight into what, either phenomenologically or ontologically, constitutes destructive plasticity (be it push, pull, or, as per plasticity’s annihilative potential, shock and blast), and of what its constitution may actually entail. This is a problem because, in not accounting for how plasticity may be destructive or creative, the concept of plasticity is somewhat disconnected from the very dynamic processes (that ultimately give and receive form) active in change. Plasticity, although I believe is  unquestionably integral to being, reads as abstracted from quite possibly non-plastic  transformative impetuses of life and world.”  

(This is later reflected, for example, in Section One on lines 438 - 443. Please see:  “On the flipside, Malabou however, appears resolved to delimit her thinking the constitution of plasticity by the self-affirming economy of its malleability. That as it may be, how Derrida wrestles with the indeterminacy of systems (that are within a broad ecology of relations) as engendering the figure of force may provide us with insight into how we begin to engage the economy of forces that are active as plasticity’s giving and receiving.”)

A clear explanation of the relation between form and plasticity is needed. The first sentence in the abstract is very confusing:

"Catherine Malabou conceives of plasticity as a creative and destructive form…"

– is plasticity a form? In a sense yes, but in a different sense, no, not really. The relation between form and plasticity is better handled later in the text, but clarification is needed. Plasticity at the Dusk of Writing (esp. chapters VIII, X and XI) and "Grammatology and Plasticity" could be helpful references here.

Sentence now reads (lines 11 to 13): "Catherine Malabou’s conception of plasticity as potentially having  a creative or destructive form  provides both philosophy and the neurosciences with a dynamic and generative concept for describing the workings and transformations of psychological, social and material phenomena. "

As per the changes to lines 11 through to 13,  I hope I’ve addressed the confusion over the relation of plasticity to form that the opening sentence of the abstract created.

Some signposting along the way would make reading easier. 

I’ve attempted to address the concerns for the need to provide the reader with a bit more indication of the direction the various parts of my analysis take by adding to the following: 

(lines 144) “ to put in terms that theorist Christopher Watkin (to whom I will later turn to for examining the necessity of the figure of force for plasticity)”

(lines 287 - 290) “Plasticity, along with, as we will explore, her later and, as I will discuss, very important turn to epigenesis, potentially provide a constructive conceptual model that dynamically accounts for the fluid and transformative interplay of the structural and the contingent, the biological and the social.

(lines 464 - 465) “requires us to now transition from examining how destruction is integral to things to exploring

Some paragraphs are much too long. Editing for readability should be done throughout.

I’ve attempted to address the concerns over unwieldy paragraph lengths by making new paragraphs at the following: 

Line 73 - new paragraph

Line 116 - new paragraph

Line 307 - new paragraph

Line 373 - new paragraph 

Line 411 -  new paragraph and now reads: “As well as how we may think about socio-political formations, there are certainly more conceptual parallels between deconstruction and plasticity. ”  

Line 433 - new paragraph

Line 504 - new paragraph

Line 584 - new paragraph

Line 648 - new paragraph

This manuscript is a resubmission of an earlier submission. The following is a list of the peer review reports and author responses from that submission.

Round 1

Reviewer 1 Report

Overall, this paper is quite promising but needs the engagement with Malabou's articulation of destructive plasticity to be strengthened to make the paper follow through on that promise. The author is right to consider that there are problematic elements to Malabou's attempt to combine destructive plasticity with the other forms of plasticity she elaborates (cf. Malabou on Canguillhem). The Hope article cited essentially argues that the problem with this is that Malabou attempts to force a (possibly Hegelian) unity out of a concept (as Deleuze would use the term) that tends towards the rhizomatic rather than the aboreal. Thus, the issue with ignoring other senses of plassein relates to this aufheben/sublation rather than that selective etymology is always problematic per se. (As a minor referencing point, it should be acknowledged that the Barthes passage is also cited in the Hope article and in Plasticity at the Dusk of Writing on a page cited in the conclusion).

The most well-developed analyses of destructive plasticity in Malabou’s work appear in The New Wounded and The Ontology of the Accident rather than being dependent on the slightly trite etymological move that Malabou makes with plastique. The author does cite a couple of passages, but perhaps needs to do more to show that ‘annihilation’ is a consistent feature of Malabou’s elaboration of destructive plasticity a little more clearly. I think the key problem is with the choice to stay on the level of a general concept in relation to neurobiology, while Malabou attempts to shore up the existence of a destructive form of plasticity and its integration into her overall conceptual framework through specific examples of destructive plasticity as ‘cerebrality’ in The New Wounded and in Plasticity at the Dusk of Writing. These examples are better worked through than the headline concept but are nonetheless problematic. I’m not sure if the examples would work with the author’s incorporation of the parasite, but it would be interesting to see.

Some good critiques of these moves in TNW and elsewhere can be found in recent books and articles on Malabou from Christopher Watkin, Emily Apter and Ian James. There also is plenty to say about the generality of plasticity as a ‘motor scheme’ for our times, and those moves to try to encompass the social in grand sweeping scope are probably some of the more problematic ones in Malabou’s work, but while Malabou herself repeatedly mentions destructive plasticity, I’m not entirely convinced it’s loss as bombing would undermine Malabou’s whole programme.

From a proof-reading perspective, there are consistent errors with the titles of books and the habit of putting dashes at the start of sentences where they have no cause to be is slightly distracting at times.

Reviewer 2 Report

At best, this paper is very uncharitable to Malabou. At worst, in many places, this paper simply misunderstands or gets Malabou wrong. The impression is that the authors jointly decided that the notion of destructive plasticity was no good but, they have failed to explain it let alone provide a coherent argument as to how or why it fails, the significance of such a failure, and what should be done about that. The paper also reads like three separate papers simply put together without consideration of the interrelation of each part or how they might each contribute to making a given argument. It would have benefitted from having a core argument to structure and guide the development of the paper and to provide a road-map to readers about what steps will be taken and why they matter to the core argument being made. As it stands, it is all over the map claiming in some places that destructive plasticity is a paradox and in other places apparently denying that there is a paradox--moreover, I don't believe Malabou's concept of destructive plasticity is at all paradoxical but no paradox was ever established in the paper. The paper contains a comparative exegetical section relating the views of Malabou and Derrida but we are never told why we should care about this or what value it has to the overall argument since the overall argument of the paper is poorly specified.

There are far too many large block quotes that are not properly formatted and often they don't meaningfully contribute to the development of an argument for the paper.

The paper seems to focus heavily upon Malabou's "Before Tomorrow" at the expense of her earlier work (viz.: "Ontology of the Accident" and "The New Wounded") with sufficient attention given to those texts, the authors may have avoided the bulk of their misunderstandings of Malabou's project and the reasons for it. For the authors not to mention that the concept of destructive plasticity was developed to understand a particular kind of subjectivity is a massive oversight as was the omission of any mention of Freud's role in the formation of Malabou's thought on the matter. 

As such, I cannot recommend that the paper be published and I don't believe it is coherent enough to be modified and resubmitted.

There are moments of insight and interesting points made here and there but they are just not enough to make up for the deep misunderstandings and general lack of coherence of the paper.

There may be a paper that could be pulled from the comparative exegetical section between Malabou and Derrida but much more would need to be done by way of developing an argument from that section for it to be viable for publication.

 The line 384-385 is probably the most interesting idea in the paper and an analysis of what it means for destruction to be productive, that is, for "the destructive to give form" might also generate thought that could add to the scholarship. 

Reviewer 3 Report

The article engages with the 'destructive' side of Catherine Malabou's concept plasticity. The article is well-written and well-structured and engages with the problem – "If the annihilative has a form or takes a form how can it then be annihilative?" – with enthusiasm. The article is provocatie and invited me to think deeply about the destructive side of plasticity. Unfortunately, the article leaves one with the sense that it does not spend its energy where it would be best.

While very well-written and generally competently handled, the article rests on the elevation of a minor metaphor in Malabou's oeuvre to a central element in her philosophy. The reference to "plastic explosives" in What should we do with our brain? is presented as the metaphor for destructive plasticity, while in the context of her oeuvre it is merely an illustrative metaphor highlighting the notion that plasticity has the potential for radical contingency ("explosion") as well as creation ("the sculpture").

In The Future of Hegel and The Heidegger Change in addition to Before Tomorrow, plasticity – including its explosive feature – presents itself as a temporal, dialectical and radically contingent feature of form, but also as form's precondition. Engaging with these works as well would have clarified how 'plasticity' and its relation to 'form' should be understood in the context of Malabou's philosophy.

Furthermore, a closer engagement with Malabou's essay on destructive plasticity, The Ontology of the Accident, would be useful. There, it is quite clear that 'explosivity' refers metaphorically to the force – not necessarily speed – with which plasticity's destructive side can manifest itself and the potential for "the Wholly other", i.e. radical contingency, to occur. In all the aforementioned works, the metamorphic character of plasticity is emphasized, whether the plasticity in question is constructive or destructive. A plasticity that explodes does so not in terms of its temporal compression but in terms of its potential for boundless change, or rather, destruction.

On this basis, the intense scrutiny on what a bomb is and how they are made, and how Malabou's illustration fails to take the chemistry of bomb-making into account, appears misguided. In fact, the approach chosen for this article presents the author with the considerable hurdle of trying to 'destroy' destructive plasticity as it is formulated by Malabou. Unfortunately, this reviewer finds this attempt unsuccessful, simply because it does not engage with what would reasonably constitute 'destructive plasticity'.

I invite the author to consider whether a different approach might be more 'constructive' – perhaps one in which the author presents their own conceptualisation of 'destructive plasticity' on the basis of some elements of Malabou's philosophy. There is no doubt that the author is capable of that.